# Learning Cross-Spectral Prior for Image Super-Resolution

## ABSTRACT

With the rising interest in multi-camera cross-spectral systems, cross-spectral images have been widely used in computer vision and image processing. Therefore, an effective super-resolution (SR) method is significant in providing high-resolution (HR) cross-spectral images for different research and applications. However, existing SR methods rarely consider utilizing cross-spectral information to assist the SR of visible images and cannot handle the complex degradation (noise, high brightness, low light) and misalignment problem in low-resolution (LR) cross-spectral images. Here, we first explore the potential of using near-infrared (NIR) image guidance for better SR, based on the observation that NIR images can preserve valuable information for recovering adequate image details. To take full advantage of the cross-spectral prior, we propose a novel **C**ross-**S**pectral **P**rior guided image **SR** approach (**CSPSR**). Concretely, we design a cross-view matching (CVM) module and a dynamic multi-modal fusion (DMF) module to enhance the spatial correlation between cross-spectral images and to bridge the multi-modal feature gap, respectively. The DMF module facilitates adaptive feature adaptation and effective information transmission through a dynamic convolution and a cross-spectral feature transfer (CSFT) unit. Extensive experiments demonstrate the effectiveness of our CSPSR, which can exploit the prominent cross-spectral information to produce state-of-the-art results.

## CCS CONCEPTS

• **Do Not Use This Code → Generate the Correct Terms for Your Paper**; *Generate the Correct Terms for Your Paper*; Generate the Correct Terms for Your Paper; Generate the Correct Terms for Your Paper.

## KEYWORDS

Cross-spectral, Super-resolution, Near-infrared, Visible image

## 1 INTRODUCTION

Nowadays, the multi-camera cross-spectral system is embedded in many modern RGBD devices, such as the RGB-NIR camera in Kinect and iPhone X, and has become increasingly

*ACM MM, 2024, Melbourne, Australia*
© 2024 Copyright held by the owner/author(s). Publication rights licensed to ACM.
ACM ISBN 978-x-xxxx-xxxx-x/YY/MM
https://doi.org/10.1145/nnnnnnn.nnnnnnn

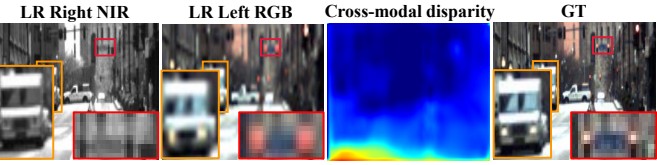

**(a) Stereo cross-spectral image**

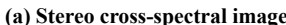

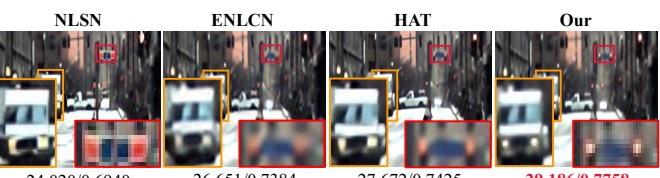

**(b) SR results of the visible image**

**Figure 1: Cross-spectral images and their SR results. (a) A pair of LR cross-spectral images (an NIR image and an RGB image in different views), the cross-spectral disparity, and the ground truth (GT) VIS image. (b) Comparison with the state-of-the-art SR methods (NLSN [59], ENLCN [47], HAT [3]) for the ×4 SR. The proposed approach can effectively use the NIR image as guidance to restore a better high-resolution VIS image with clear details and fine structure.**

popular. Cross-spectral images receive wide attention in the computer vision field and provide strong benefits for numerous practical applications, such as scene parsing [5], person re-identification [11, 29], face recognition [4, 13–15], object detection [12, 25, 39]. These applications always require high-resolution (HR) images. Therefore, the super-resolution of cross-spectral images (CSSR), producing high-resolution (HR) images from their low-resolution (LR) versions, is significant. However, real-world cross-spectral images always suffer from complex degradation, such as noise, high brightness, and low light, making the CSSR challenging. Existing SR methods (NLSN [59], ENLCN [47], HAT [3]) cannot perform well on cross-spectral images (see Figure 1).

The near-infrared (NIR) image and the visible (VIS) RGB image of the cross-spectral images in Figure 2 present different inherent characteristics. Compared with the VIS image, the NIR image retains better brightness contrast and richer texture details in some overexposed or dark areas and provides clearer boundaries of the texts, as it is robust to the change of colour and is sensitive to the change of illumination. In addition, the NIR image can resist the disturbance of bad imaging conditions, such as low illumination, and fog. Therefore, the NIR image is able to preserve some valuable information for recovering adequate VIS image details and to provide many benefits for the VIS image SR.

The above analysis inspires us to propose a cross-spectral prior guided super-resolution (CSPSR) approach by introducing the NIR guidance into the SR of the VIS image. Figure 1 (a) demonstrates a pair of LR cross-spectral images and the disparity between them. The multi-modal feature gap and cross-view pixel misalignment make the CSPSR challenging. To our knowledge, most current SR methods mainly work on images of single-modal or single-view and rarely exploit the cross-spectral image information (e.g., NIR) to guide the (VIS) image SR. Therefore, it is suboptimal to directly apply existing SR methods to the CSPSR.

How to bridge the multi-modal feature gap and enhance the cross-view sub-pixel correspondence are prominent insights of this paper. The proposed CSPSR approach can restore accurate HR images by taking full advantage of the cross-spectral and cross-view image information. Specifically, we enhance the pixel-level correspondence between different views through a cross-view matching (CVM) module to provide more appropriate NIR guidance for the VIS image. To fully fuse multi-modal features of cross-spectral images, we design a dynamic multi-modal fusion (DMF) module, composed of a dynamic convolution and a confidence-based cross-spectral feature transfer (CSFT) unit. The dynamic convolution adaptively adapts the NIR feature to better match the VIS feature, and the CSFT unit transfers reliable knowledge from the NIR image to the VIS image by learning different confidence maps.

As shown in Figure 1 (b), our CSPSR can effectively utilize the NIR information to produce an HR VIS image with clear textures and structures, that are closer to the ground truth (GT). The highlights of our work are as follows.

- We analyze the inherent characteristics of the cross-spectral images and propose a novel cross-spectral prior guided image super-resolution (CSPSR) approach, which introduces the NIR image to assist the SR of the VIS image for the first time.
- We propose a cross-view matching (CVM) module and a dynamic multi-modal fusion (DMF) module to take full advantage of the cross-spectral and cross-view image information for better SR. The CVM module enhances the cross-view spatial correspondence. The DMF module adaptively bridges the multi-modal feature gap and transfers cross-spectral knowledge through a dynamic convolution and a cross-spectral feature transfer (CSFT) unit.
- We design a dual-branch framework for extracting multi-modal features of cross-spectral images to provide an informative reference related to SR.

## 2 RELATED WORK

### 2.1 Single-Modal Image Super-Resolution

Benefiting from the development of deep learning, single-image super-resolution (SISR) has achieved remarkable advances over previous reconstruction-based methods [2, 20, 34, 54]. As the first convolutional neural network (CNN)

based SISR method, SRCNN [9] learns the LR-to-HR mapping and achieves remarkable advances. Following this fashion, a large number of deep learning-based SISR methods [3, 19, 23, 26, 36, 42, 47, 50, 52, 55–57, 59, 62, 64]) have been developed to improve the objective accuracy or perceptual quality of the SISR results.

Most SISR approaches adopt MSE or MAE as a loss function and target high PSNR/SSIM by proposing various deep network architectures. For instance, VDSR [19] constructs a deeper SISR network with 20 convolution layers. EDSR [23] builds a very deep and wide network by cascading modified residual blocks. DRFN [56] adopts a deep recurrence learning strategy to enlarge the receptive field and utilizes transposed convolution for upsampling. ENLCN [47] proposes an efficient non-local contrastive attention module to model long-range visual features and leverage more relevant non-local features in an image. HAT [3] combines both channel attention and self-attention schemes to utilize global image statistics. To decrease the computational cost, NLSN [59] proposes a Non-Local Sparse Attention (NLSA) and embeds it into a residual network to enforce sparsity in the Non-Local attention module, as well as largely reduce its computational cost. SMSR [42] learns sparse spatial and channel masks to identify important locations and mark redundant channels in those unimportant regions for efficient SR.

To obtain better SR performance, some methods use additional images prior to assisting the super-resolution. Specifically, multi-frame super-resolution [31], reference-based super-resolution (RefSR) [27, 48, 53, 60, 65], and stereo image super-resolution [7, 10, 17, 24, 43, 51, 68] methods exploit images from adjacent frames, reference images and cross-view images, respectively, to provide beneficial guidance for SR and achieve a huge breakthrough.

### 2.2 Multi-Modal Image Restoration

In addition to single-modal visible images, multi-modal images, including infrared images and near-infrared (NIR) images, have also been regarded as a prior in some RGB image restoration tasks, such as the low-light image enhancement [33, 63, 69], image dehazing [21, 38], image restoration [16, 46], and image denoising [44].

For instance, some image enhancement methods [33, 63, 69] use the contrast and texture information in infrared images to guide the enhancement of low-light VIS images. The NIR image-guided colour image denoising method [44] fuses NIR images and noisy colour images to eliminate image noise and transfer detail structure from guided images by simply concatenating the two input images.

In order to utilize the multi-modal images in SR, most current multi-modal image super-resolution (MMSR) methods regard RGB images as the guidance for the SR of images from other modalities, including depth image [8, 30, 35, 37, 40, 41, 45, 49, 58, 70], multi-spectral image [22, 28, 61], thermal image [35, 49], NIR image [35]. Some self-supervised MMSR methods [30, 35, 49] only require LR source and HR RGB images for training by constructing pseudo supervision in

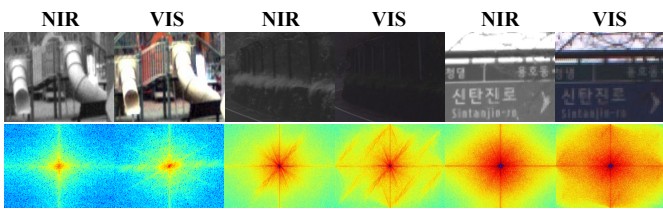

(a) Frequency difference between the cross-spectral images.

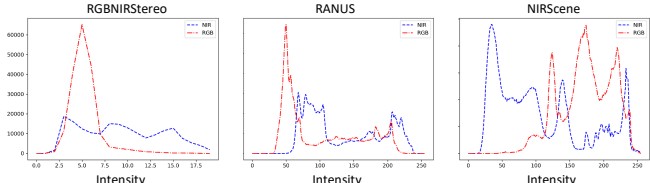

(b) Pixel intensity distribution of NIR-VIS images in three testsets.

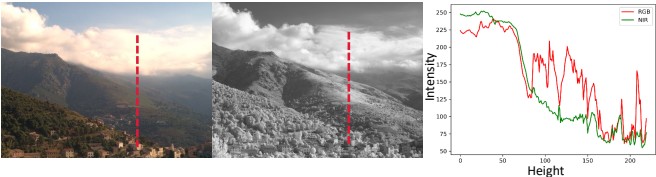

(c) Intensity fluctuation of pixels on the red dotted line.

**Figure 2: Visual and statistic comparison between cross-spectral images (NIR and VIS). (a) The spectrograms are shown below the NIR-VIS images. Some texture details, that are blurred in the VIS images, are clear and sharp in the NIR images. (b) The pixel intensity distribution of NIR-VIS images in RGB-NIR stereo [66], NIRScene [1], and RANUS [6] datasets. (c) Intensity fluctuation of pixels along the red dotted lines on the left NIR-VIS images.**

LR space or using the weakly supervised cross-modal transformation manner. CMSR [49] is proposed to super-resolve the thermal, NIR, and depth images under the guidance of RGB images. However, these RGB-guided MMSR methods require high-resolution RGB images as reference, which are unavailable in some real-world scenes.

Current MMSR methods ignore the potential of the NIR image for improving the VIS image quality and require pixel-level aligned multi-modal images. NIR images have not been considered to assist the SR of visible (RGB) images. In addition, their practicability is limited when the HR RGB image is unavailable.

## 3 NIR-VIS IMAGE ANALYSIS

To effectively introduce the NIR guidance into the VIS image SR, we first analyze the inherent characteristics of the NIR-VIS images. Figure 2 demonstrates the visual and statistical characteristics of the NIR and VIS images.

As shown in Figure 2 (a), the NIR images retain better brightness contrast and richer texture details under poor lighting conditions, while the VIS images suffer from detail loss in the overexposed or dark areas. Furthermore, the NIR

images provide simpler and clearer boundaries of texts, as they are more sensitive to illumination change. In comparison, the VIS images need to handle complex colours, leading to blurred or jagged edges between different objects. In addition, VIS images are susceptible to illumination, fog, and other bad weather, and NIR images can well resist these disturbances. Furthermore, we also observe that the NIR images preserve accurate high-frequency information, which is helpful in providing reasonable guidance and few low-frequency disturbances for recovering adequate image details.

Then, we also show the distribution of average pixel intensity in NIR/VIS images from three benchmark datasets (RGB-NIR stereo [66], NIRScene [1], and RANUS [6]) in Figure 2 (b). Compared to VIS images, the pixel intensity distribution of NIR images is relatively more uniform, demonstrating that NIR images can provide beneficial and complementary guidance for recovering accurate and rich VIS image details. Figure 2 (c) visualizes the intensity variation of pixels along the red dotted lines in the left NIR/VIS image. The NIR image presents a larger step response between mountains and clouds, therefore, it retains clearer boundaries. In addition, for the rich-textured VIS image areas with a high-frequency intensity variation, the NIR image has weaker intensity changes, which ensures more valuable NIR information is provided and avoids the disturbance to details that already exist in the VIS image.

Based on the above analysis, the visual and statistical difference between the NIR-VIS images inspires us to take full advantage of the beneficial information from the NIR image for guiding the SR of the VIS image.

## 4 METHOD

To take full advantage of the cross-spectral guidance for better SR, we propose a cross-spectral prior guided image super-resolution (CSPSR) approach. We will dedicate to stating the proposed approach in detail in the following subsections.

### 4.1 Overview

Figure 3 demonstrates the overall workflow of the CSPSR framework. We adopt a dual-branch structure to super-resolve images of different modalities, as the two SR branches can extract valuable features, that are more relevant to SR, to provide more appropriate guidance. To bridge the multi-modal feature gap and enhance the cross-view sub-pixel correspondence of cross-spectral images, we design a cross-view matching (CVM) module and a dynamic multi-modal fusion (DMF) module, which can enhance the correspondence between two views and fuse features of different modalities, respectively. We will introduce the two modules in detail in the following subsections.

As shown in Figure 3, given a pair of LR cross-spectral images, composed of a left VIS image ($I_{LR}^l \in R^{H \times W \times 3}$) and a right NIR image ($NIR_{LR}^r \in R^{H \times W \times 1}$), the CSPSR model first aligns the two images to obtain an NIR image ($NIR^l$) in the left view through the CVM module. Then, we deliver the $I_{LR}^l$ and $NIR^l$ into two SR branches. Concretely, each

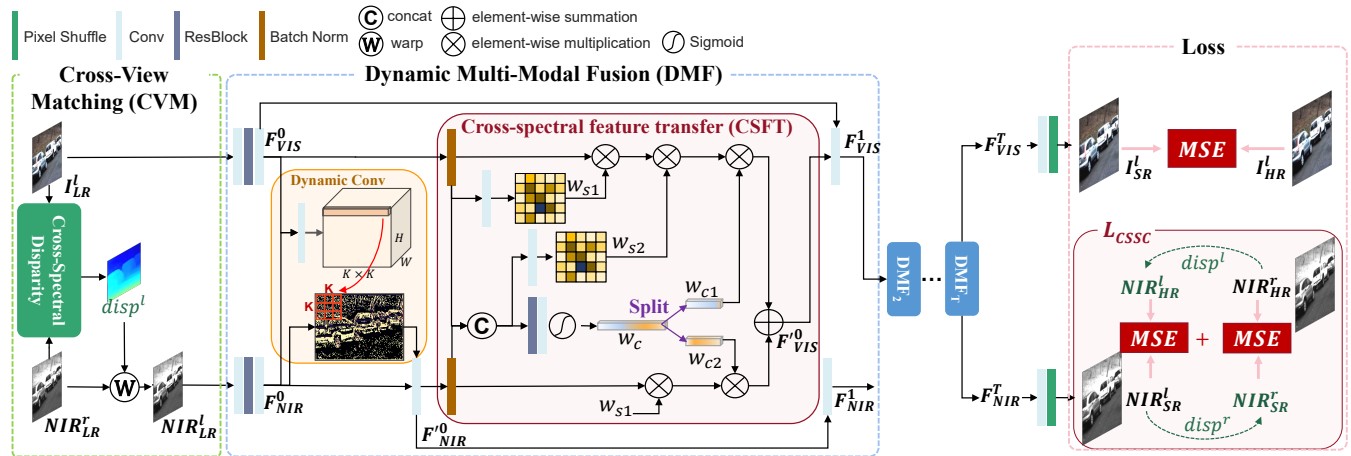

**Figure 3: Overview of the proposed cross-spectral prior guided image super-resolution (CSPSR) framework. Given a pair of LR VIS image ($I_{LR}^l$) and LR NIR image ($NIR_{LR}^r$), the CSPSR adopts a cross-view matching (CVM) module to enhance the correlation between the cross-spectral input images, generating the NIR image in the left view ($NIR^l$). Then, the dual-branch network extracts the middle SR features ($F_{VIS,t=1:T}^{t-1}, F_{NIR,t=1:T}^{t-1}$) from the matched cross-spectral images and fuses the features of two modalities adaptively by repeating the dynamic multi-modal fusion (DMF) module $T$ times. The cross-spectral feature transfer (CSFT) unit adaptively transfers information from the $F_{NIR}^{\prime t-1}$ to the VIS feature space, yielding $F_{VIS}^{\prime t-1}$. The final SR results ($I_{SR}^l, NIR_{SR}^l$) are reconstructed through a convolution layer and a pixel shuffle layer. The total loss of the CSPSR is composed of the MSE on the SR result of the VIS image and the cross-spectral spatial consistency loss ($\mathcal{L}_{CSSC}$).**

SR branch first extracts the shallow features ($F_{VIS}^0, F_{NIR}^0 \in R^{H \times W \times C}$) of the inputs ($I_{LR}^l, NIR^l$) with two convolution layers and a residual block. The residual block is composed of two convolution layers and a residual connection. $H, W, C$ denote the height, width, and channel number of the features ($F_{NIR}, F_{VIS}$).

To adaptively fuse the NIR and VIS image features, the DMF module, consisting of a dynamic convolution operation and a cross-spectral feature transfer (CSFT) unit, is inserted between two SR branches. The DMF module takes the NIR image feature ($F_{NIR,t=1:T}^{t-1}$) and the VIS image feature ($F_{VIS,t=1:T}^{t-1}$) as input and is repeated for $T$ times to fully combine the NIR and VIS image features. After $T$ DMF modules, the outputs ($F_{VIS}^T, F_{NIR}^T$) of the last DMF module in each SR branch are delivered to a convolution layer and a pixel shuffle layer to generate the final SR results ($I_{SR}^l, NIR_{SR}^l$).

## 4.2 Cross-View Matching (CVM) Module

As shown in Figure 3, the CVM module enhances the global correspondence between the cross-spectral images based on the cross-spectral disparity prediction model [66]. In order to enhance the robustness of the CSPSR model, we predict rough left-right disparities $\{disp^l, disp^r\}$ based on the low-frequency LR images directly. Thus, we can also allocate more training effort to the subsequent SR modules by decreasing difficulty and releasing the burden of training the CVM module.

Then, we offset all pixels $(i, j)$ in the right NIR image according to the $disp^l$, which can accurately align cross-spectral images to the same view, and generate the new NIR image in the left view ($NIR^l$), facilitating stronger sub-pixel correspondence between the NIR/VIS images.

$$NIR^l(i, j) = NIR^r(i, j - disp^l(i, j)) \qquad (1)$$

Figure 4 gives an example of the output of the CVM module. By enhancing the pixel-level correlation between cross-spectral images, the convolutional calculation with a limited receptive field can take full advantage of the aligned NIR image to restore more accurate VIS image details.

## 4.3 Dynamic Multi-Modal Fusion (DMF) Module

As shown in Figure 3, the $t$-th dynamic multi-modal fusion (DMF) module first adopts a dynamic convolution to adapt the $F_{NIR}^{t-1}$, resulting in a new NIR image feature ($F_{NIR}^{\prime t-1}$). Then, to transfer knowledge from the NIR image to the VIS image, we feed the $F_{VIS}^{t-1}$ and the $F_{NIR}^{\prime t-1}$ to the CSFT unit and obtain a new VIS image feature ($F_{VIS}^{\prime t-1}$), containing the information in the NIR image feature. Finally, the input features ($F_{NIR}^t, F_{VIS}^t$) of the next DMF module ($DMF_{t+1}$) are obtained by two convolution layers.

**Dynamic convolution.** Inspired by the dynamic upsampling filter [18], we introduce a dynamic convolution to conduct content-adaptive feature adaption. First, we concatenate the $F_{VIS}^{t-1}$ and the $F_{NIR}^{t-1}$ and apply a convolution layer to learn

**Figure 4: The outputs of the cross-view matching (CVM) module, which takes the LR left VIS image ($I_{LR}^l$) and right NIR image ($NIR_{LR}^r$) as input and outputs the left NIR image and the disparity ($NIR^l, disp$). The pixel $(i, j)$ in $I_{LR}^l$ corresponds to the pixel $(i, j - d(i, j))$ in $NIR_{LR}^r$, where $d(i, j)$ denotes the corresponding value in the $disp^l$.**

a kernel map ($K \in R^{H \times W \times k^2}$). Then, we reshape all vectors $K(i, j)_{i=1:H, j=1:W} \in R^{1 \times 1 \times k^2}$ to generate $H \times W$ filters of size $R^{k \times k}$. Finally, the new NIR image feature ($F_{NIR}^{\prime t-1}$) can be obtained by position-aware filtering the $F_{NIR}^{t-1}$ with the $K$. As the kernel is learned based on the NIR-VIS features, it can flexibly adapt the NIR feature and make it more compatible to enhance the VIS feature.

**Cross-spectral feature transfer (CSFT) unit.** The aligned images can provide more relevant information, which makes it easier to guide the SR of the left VIS image. However, due to the issue of occlusions, the CVM module cannot guarantee sub-pixel matching accuracy, which may degrade the SR performance and introduce unwanted artefacts. Therefore, how to effectively extract and utilize helpful information in the aligned cross-spectral images is significant. In addition, fusing features of different modalities adaptively is also a key point that needs to be solved.

To address the above problems and make full use of the NIR image for better SR, we propose a confidence-based CSFT unit, where the multi-modal features are weighted for better feature fusion between cross-spectral images. As shown in Figure 3, the CSFT unit can adaptively fuse the NIR-VIS image features ($F_{NIR}^{\prime t-1}, F_{VIS}^{t-1}$) to output a new feature ($F_{VIS}^{\prime t-1}$) through learnable spatial and channel-wise attention. Thus, the feature $F_{VIS}^{t-1}$, combining abundant and useful information of the NIR image and VIS image, is helpful for restoring more texture details in the VIS image. The specific workflow of the CSFT unit is as follows.

As we mentioned before, some VIS image regions with low contrast typically lose many details. From this observation, we first learn a spatial weight ($w_{s1} \in R^{H \times W \times 1}$) based on the $F_{VIS}^{t-1}$ through a convolution layer with kernel size $3 \times 3 \times 1$ to indicate image areas with poor quality. Considering that cross-spectral images have different intensity ranges and

visual effects, we apply two batch normalization (BN) layers on $F_{NIR}^{\prime t-1}, F_{VIS}^{t-1}$ to unify the NIR-VIS image features and highlight the relative difference. Then, to rebalance features of different modalities, we also learn a channel weight ($w_c \in R^{1 \times 1 \times 2C}$) by passing the $F_{NIR}^{\prime t-1}$ and $F_{VIS}^{t-1}$ to a global average pooling layer, a convolution layer with kernel size $1 \times 1 \times 2C$, and a sigmoid activation layer, respectively. Next, $w_c$ is split into two vectors ($w_{c1}, w_{c2} \in R^{1 \times 1 \times C}$) to re-weight the $F_{NIR}^{\prime t-1}, F_{VIS}^{t-1}$ based on their channel-wise correlation and significance.

Given the concatenation of $F_{NIR}^{\prime t-1}$ and $F_{VIS}^{t-1}$, another spatial weight map ($w_{s2}$) is generated through a convolution layer with kernel size $3 \times 3 \times 1$ to identify the accurate and useful NIR image features. Finally, we combine the $F_{NIR}^{\prime t-1}$ and $F_{VIS}^{t-1}$ by weighted summation based on the weights $w_{s1}, w_{s2}, w_{c1}, w_{c2}$ to transfer the abundant information from the NIR image to the VIS image.

$$F_{VIS}^{\prime t-1} = F_{NIR}^{\prime t-1} \times w_{s1} \times w_{c2} + F_{VIS}^{t-1} \times w_{s1} \times w_{s2} \times w_{c1},$$
$$(2)$$

where $\times$ denotes the element-wise multiplication.

## 4.4 Cross-Spectral Spatial Consistency Loss

Given the VIS-NIR outputs ($I_{SR}^l, NIR_{SR}^l$) and the GT cross-spectral images ($I_{HR}^l, NIR_{HR}^r$), our CSPSR model is trained end-to-end using the final loss ($\mathcal{L}$) in Eq.(3). In addition to the mean square error (MSE) loss, used to constrain the pixel-level accuracy between the $I_{SR}^l$ and $I_{HR}^l$, we also propose a cross-spectral spatial consistency loss ($\mathcal{L}_{CSSC}$) for optimizing the network to learn better NIR information. Since calculating the MSE between the misaligned NIR images ($NIR_{SR}^l, NIR_{HR}^r$) directly will lead to spatial artefacts in the NIR image, we first obtain the HR left NIR image ($NIR_{HR}^l$) and the SR right NIR image ($NIR_{SR}^r$) by warping the $NIR_{HR}^r$ and the $NIR_{SR}^l$ based on the disparity ($disp^l, disp^r$) (Eq.(1)). Then, the $\mathcal{L}_{CSSC}$ calculates the MSE between the aligned NIR image pairs, including ($NIR_{SR}^l, NIR_{HR}^l$) and ($NIR_{SR}^r, NIR_{HR}^r$).

$$\mathcal{L} = |I_{SR}^l - I_{HR}^l|_2 + \lambda_1 \mathcal{L}_{CSSC},$$
$$\mathcal{L}_{CSSC} = |NIR_{SR}^l - NIR_{HR}^l|_2 + |NIR_{HR}^r - NIR_{SR}^r|_2,$$
$$(3)$$

where the weight $\lambda_1$ is set to 0.1.

## 4.5 Implementation Details

The final CSPSR model contains 15 DMF modules ($T = 15$) in total. The kernel size of the convolution layers in two SR branches is $3 \times 3$ and the feature number $C$ is 64.

## 5 EXPERIMENTS

This section mainly introduces experimental settings and reports the performance of our approach by conducting the

**Table 1: Quantitative comparison with state-of-the-art SR approaches. The average PSNR↑/SSIM↑ of VIS images on three NIR-VIS datasets (RGB-NIR stereo [66], NIRScene [1], RANUS [6]). ↑ denotes the higher, the better. The best results are highlighted in bold. The GFLOPs/Param(MB) denote the calculation amounts and the parameter amounts of different SR models.**

| Dataset | scale | Bicubic | EDSR | RCAN | SMSR | NLSN | ENLCN | HAT | Our |
|---|---|---|---|---|---|---|---|---|---|
| RGB-NIR stereo | ×2 | 26.484/0.7664 | 29.592/0.8153 | 30.959/0.8179 | 30.176/0.8501 | 30.296/0.8531 | 30.288/0.8527 | 30.777/0.8628 | **32.212/0.8584** |
| | ×4 | 22.056/0.5628 | 24.928/0.6841 | 25.245/0.6915 | 25.471/0.7018 | 25.377/0.6971 | 25.440/0.6999 | 25.560/0.7195 | **27.765/0.7494** |
| NIRScene | ×2 | 32.455/0.9075 | 33.915/0.9259 | 33.937/0.9265 | 33.486/0.9191 | 33.966/0.9295 | 34.012/0.9325 | 34.506/0.9306 | **35.177/0.9410** |
| | ×4 | 28.276/0.7784 | 30.293/0.8339 | 29.802/0.7963 | 29.265/0.7801 | 29.641/0.7841 | 29.905/0.8002 | 29.955/0.8065 | **30.856/0.8454** |
| RANUS | ×2 | 39.169/0.9716 | 41.059/0.9703 | 41.350/0.9725 | 41.453/0.9733 | 42.004/0.9774 | 42.038/0.9778 | 42.338/0.9782 | **43.466/0.9885** |
| | ×4 | 34.815/0.9092 | 35.992/0.9115 | 36.335/0.9204 | 35.895/0.9097 | 36.307/0.9280 | 36.364/0.9305 | 36.608/0.9328 | **37.501/0.9380** |
| GFLOPs/Param(MB) | - | | 246.586/1.518 | 247.592/15.592 | 2.710/1.005 | 36.030/1.853 | 86.248/1.536 | 65.758/9.211 | 51.059/1.298 |

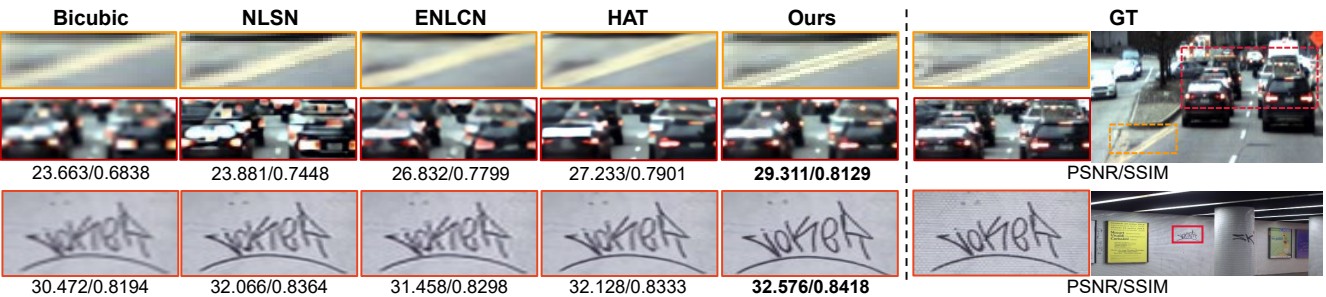

| Bicubic | NLSN | ENLCN | HAT | Ours | GT |
|---|---|---|---|---|---|
| 23.663/0.6838 | 23.881/0.7448 | 26.832/0.7799 | 27.233/0.7901 | **29.311/0.8129** | PSNR/SSIM |
| 30.472/0.8194 | 32.066/0.8364 | 31.458/0.8298 | 32.128/0.8333 | **32.576/0.8418** | PSNR/SSIM |

**Figure 5: Visual comparison with the state-of-the-arts (NLSN [59], ENLCN [47], HAT [3]). The ×4 VIS SR results on the RGB-NIR stereo [66] and NIRScene [1] datasets.**

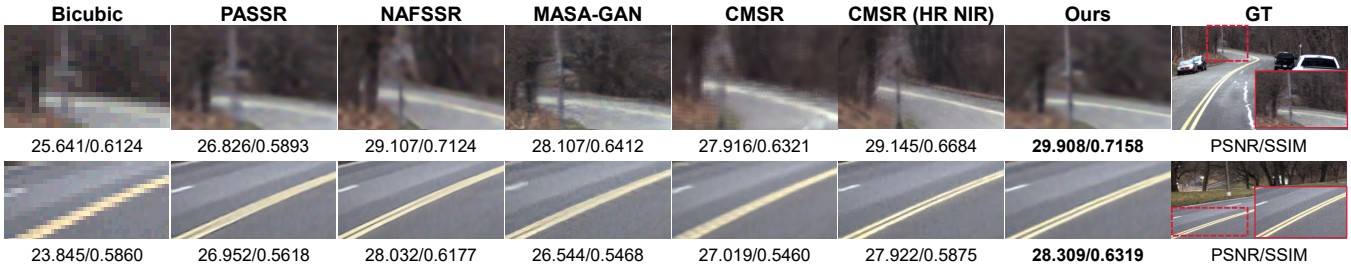

| Bicubic | PASSR | NAFSSR | MASA-GAN | CMSR | CMSR (HR NIR) | Ours | GT |
|---|---|---|---|---|---|---|---|
| 25.641/0.6124 | 26.826/0.5893 | 29.107/0.7124 | 28.107/0.6412 | 27.916/0.6321 | 29.145/0.6684 | **29.908/0.7158** | PSNR/SSIM |
| 23.845/0.5860 | 26.952/0.5618 | 28.032/0.6177 | 26.544/0.5468 | 27.019/0.5460 | 27.922/0.5875 | **28.309/0.6319** | PSNR/SSIM |

**Figure 6: Visual comparison with the prior-guided SR methods, including StereoSR (NAFSSR [7]), reference-based SR (MASA [27]), and multi-modal SR (CMSR [49]) methods.**

comparison experiment and the ablation experiment. Furthermore, the cross-spectral disparity prediction results are provided to verify the practicability of the CSPSR.

## 5.1 Dataset and Protocols

We train our models on the RGB-NIR stereo dataset [66], which contains 12 videos in total. The RGB-NIR stereo dataset contains 8 training videos and 4 testing videos, where each video covers 500 pairs of RGB-NIR cross-view frames with spatial size 582×429. We employ the RGB-NIR stereo, the NIRScene [1] (477 pairs of NIR-VIS images), and the RANUS [6] (40k pairs of NIR-VIS images) datasets for evaluation. Specifically, images in four testing videos of the RGB-NIR stereo dataset, 46 pairs of NIR-VIS images, corresponding to 9 categories, from the NIRScene dataset, and

the $50th$ subset (104 NIR-VIS image pairs) in the RANUS dataset are adopted to test different SR models.

During training, each mini-batch contains 32 pairs of cross-spectral image patches of size 120×120. The LR NIR/VIS images are generated by bicubic interpolation with scales 2, and 4. We augment the training data by randomly downscaling, flipping, and rotating images. To measure the SR results, we adopt the peak signal-to-noise ratio (PSNR) and structure similarity (SSIM) [67]. The higher PSNR/SSIM indicates better performance.

All models are based on the Pytorch implementation and optimized by Adam [32] with $\beta1 = 0.9$, $\beta2 = 0.999$. All experiments are conducted on an Nvidia GTX2080Ti GPU (128G RAM). Our model is optimized for 300 epochs with a learning rate of 1e-4.

**Table 2: Quantitative comparison with prior-guided SR methods, including StereoSR (PASSR [43], NAFSSR [7]), reference-based SR (MASA [27]), and multi-modal SR (CMSR [49]) models.**

| Testset | PASSR | NAFSSR | MASA | CMSR | CMSR (HR_NIR) | Our |
|---|---|---|---|---|---|---|
| RGB-NIR | 26.751/0.7214 | 27.529/0.7326 | 25.965/0.6677 | 26.04/0.6727 | 27.815/0.7411 | **27.765/0.7494** |
| RANUS | 36.579/0.9710 | 36.953/0.9352 | 37.433/0.9341 | 36.964/0.9094 | 37.889/0.9479 | **37.501/0.9380** |
| nirscene | 29.900/0.8981 | 30.562/0.8391 | 29.867/0.7984 | 29.299/0.7906 | 30.993/0.8479 | **30.856/0.8454** |

## 5.2 Comparison with Prior Art

**SISR methods.** To our best knowledge, existing SR methods have never explored the NIR guidance for the SR of VIS images. In order to compare with the state-of-the-art, we train and evaluate several representative SISR models, including EDSR [23], RCAN [64], SMSR [42], NLSN [59], ENLCN [47], HAT [3], on the VIS images of the RGB-NIR stereo dataset.

Table 1 reports the quantitative SR results (PSNR/SSIM) of the VIS images on three datasets and also compares the calculation amounts (GFLOPs) and parameter amounts of different SR models. Our CSPSR substantially achieves the best PSNR/SSIM with acceptable calculation amount and parameter amount. Figure 5 demonstrates that the proposed CSPSR is able to produce photo-realistic images with accurate structure and clear textures, resulting in satisfactory visual performance.

**Prior-guided SR methods.** To further verify whether the existing prior-guided SR methods work well on our problem, we retrain the StereoSR (PASSR [43], NAFSSR [7]), the reference-based SR (MASA [27]), and the multi-modal SR (CMSR [49]) models on our training data. Specifically, we replace the stereo VIS images with the NIR-VIS images as the input for the StereoSR model. Since our model only requires LR NIR image guidance, we take the Bicubic upscaled NIR image instead of the HR VIS image as the reference to guide the SR of the VIS image to the MASA and the CMSR models for fair comparison. We also construct the CMSR (w/ HR NIR) by taking the HR NIR image as guidance, which reflects the possible upper bound of the NIR-guided VIS image SR. Note that, the NIR-VIS images are first aligned through the cross-spectral stereo matching model [66] before SR.

Table 2 and Figure 6 demonstrate the quantitative and visual comparison on ×4 SR. Though our CSPSR only leverages LR NIR image, its SR results are comparable with that of the HR NIR image-guided CMSR (w/ HR NIR), which indicates the effectiveness of our strategy to exploit cross-spectral and cross-view information for SR.

## 5.3 Ablation Study

To verify the effectiveness of the proposed module for using NIR image information, we retrain a SISR model [47] (ENLCN w/ NIR), which takes the concatenation of RGB and NIR images as input. Table 3 reports the ×4 SR results on the RGB-NIR stereo dataset [66]. We can observe a slight

**Table 3: SR results on four testing videos in the RGB-NIR stereo dataset [66]. Comparison with the original ENLCN model and modified 'ENLCN w/ NIR', which takes the concatenation of NIR and VIS image as the input.**

| | 0224_0742 | 0222_0951 | 0222_1423 | 0223_1639 | mean |
|---|---|---|---|---|---|
| ENLCN w/ NIR | 25.874/0.6568 | 26.884/0.7761 | 25.637/0.7720 | 24.003/0.6229 | 25.600/0.7070 |
| ENLCN | 25.704/0.6491 | 26.600/0.7658 | 25.384/0.7605 | 23.821/0.6130 | 25.440/0.6999 |
| Ours | 27.615/0.6980 | 29.525/0.8188 | 28.210/0.8178 | 25.708/0.6630 | 27.765/0.7494 |

**Table 4: Ablation study on the different components. Average PSNR↑/SSIM↑ of ×4 SR results.**

| | | 0224_0742 | 0222_0951 | 0222_1423 | 0223_1639 |
|---|---|---|---|---|---|
| Single Branch | *Baseline* | 26.279/0.6407 | 28.315/0.7934 | 26.901/0.7898 | 23.969/0.5942 |
| | *w/ NIR* | 27.160/0.6641 | 29.484/0.8107 | 28.014/0.8089 | 24.937/0.6166 |
| | *w/ CVM* | 26.974/0.6578 | 29.448/0.8101 | 27.978/0.8081 | 24.762/0.6128 |
| Dual Branch | *w/o CVM* | 27.061/0.6646 | 26.232/0.7748 | 24.430/0.7638 | 21.036/0.5529 |
| | *w/o CSFT* | 27.162/0.6660 | 29.561/0.8122 | 28.117/0.8111 | 24.993/0.6200 |
| | *w/o DyConv* | 27.377/0.6925 | 29.075/0.8124 | 27.759/0.8105 | 25.451/0.6583 |
| | *w/o $\mathcal{L}_{CSSC}$* | 27.336/0.6821 | 29.223/0.8163 | 27.993/0.8142 | 25.653/0.6598 |
| | *Full* | **27.615/0.6980** | **29.525/0.8188** | **28.210/0.8178** | **25.708/0.6630** |

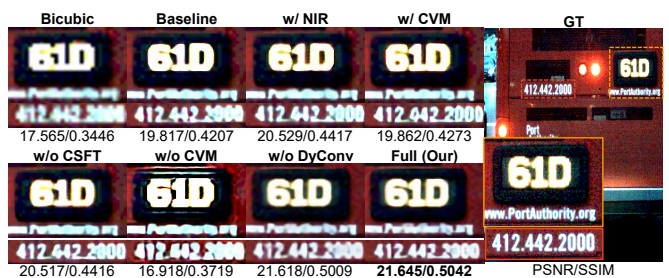

**Figure 7: Visual SR results of the ablation study. Compared with the 'w/o CSFT', which generates blurred textures, and the 'w/o CVM', which produces some artefacts, our final model 'Full' achieves the best visual results with fine details.**

improvement in SR accuracy after feeding NIR images into the SISR model, which proves that simply concatenating NIR and RGB images does not effectively exploit the valuable information in NIR images.

To verify the contribution of each component in our C-SPSR, we construct multiple SR models, three of which were single-branch and the other were double-branch, with different design options and indicate the quantitative SR results (×4) on 4 testing videos ('0224_0742', '0222_0951', '0222_1423', '0223_1639') from the RGB-NIR stereo dataset in Table 4.

First, we construct three single-branch models ('*Baseline*', '*w/ NIR*', and '*w/ CVM*'). The '*Baseline*' adopts the structure of the VIS image SR branch to learn the LR-to-HR VIS image mapping directly. Based on the '*Baseline*', the '*w/ NIR*' takes the concatenation of the LR VIS image and LR NIR image as input to introduce the NIR guidance. The PSNR/SSIM improvement of the '*w/ NIR*' over the '*Baseline*'

**Table 5: Cross-spectral disparity estimation results on SR images. Disparity RMSE ↓ in pixels for different materials.**

| Method | Common | Light | Glass | Glossy | Vegetation | Skin | Clothing | Mean |
|---|---|---|---|---|---|---|---|---|
| Bicubic | 1.3565 | 1.701 | 1.9886 | 2.3736 | 1.6742 | 1.5242 | 2.0174 | 1.5795 |
| EDSR | 0.7995 | 0.8368 | 1.0887 | 1.6819 | 0.9763 | 1.7356 | 0.7425 | 0.9827 |
| RCAN | 0.7978 | 0.8260 | 1.0875 | 1.6756 | 0.9731 | 1.7267 | 0.7374 | 0.9780 |
| SMSR | 0.8001 | 0.8541 | 1.0939 | 1.6775 | 0.9772 | 1.7522 | 0.7398 | 0.9868 |
| NLSN | 0.7999 | 0.8506 | 1.0893 | 1.6911 | 0.9753 | 1.7410 | 0.7447 | 0.9865 |
| ENLCN | 0.8610 | 1.0687 | 0.7465 | 1.0737 | 1.0338 | 1.5752 | 1.1080 | 1.1087 |
| Our | **0.5210** | **0.2558** | **0.3899** | **0.5940** | **0.6113** | **0.9846** | **0.5131** | **0.4837** |
| HR | 0.5109 | 0.2930 | 0.3912 | 0.3965 | 0.5973 | 1.0353 | 0.5700 | 0.4743 |

demonstrates that the NIR image can provide positive guidance for restoring accurate VIS image details, though the 'w/ NIR' uses the unmatched NIR-VIS images. Then, based on the 'w/ NIR', the 'w/ CVM' inserts the CVM module to align the NIR image and the VIS image before concatenating them. The CVM module may produce structure distortion in the warped NIR image, and the 'w/ CVM' fails to tolerate such error in the NIR image. Therefore, directly concatenating the aligned NIR and VIS image slightly decreases the PSNR.

We use 'Full' to denote the complete model, based on which the 'w/o CVM' and the 'w/o CSFT' remove the CVM module and the CSFT unit, respectively. The 'w/o CVM' fuses NIR-VIS image features by directly concatenating them. The comparison verifies that the CVM module enables the NIR image to provide a more valuable reference for recovering image details and the CSFT module is helpful for the model to better fuse and utilize multi-modal features for the VIS image SR. The 'w/o DyConv' replaces the dynamic convolution with the common convolution.

The 'w/o $\mathcal{L}_{CSSC}$' only calculates the MSE between the SR results and the GT of VIS images and ignores the explicit constraint for the NIR SR branch and the CVM module, obtaining lower accuracy. In conclusion, the 'Full' can adaptively exploit valuable NIR image information, leading to the highest PSNR/SSIM. In Figure 7, the SR results of the 'Full' contain finer textures that are sharper and closer to the GT, compared with that of the 'w/o CVM' and 'w/o CSFT'.

All the above experiments demonstrate the reasonability and effectiveness of the proposed network architecture and modules, which facilitate better SR by fully using the guidance of the cross-view NIR image.

Table 5 reports the disparity RMSE, calculated on different SR results on the video '0224_0742' of the RGB-NIR stereo dataset [66]. Figure 8 visualizes the cross-spectral disparity. This experiment demonstrates that our CSPSR provides high-quality cross-spectral images for the downstream task.

### 5.4 Cross-Spectral Disparity Prediction

To further demonstrate the practical value of the CSPSR, we estimate the cross-spectral disparity between the SR VIS image and the HR NIR image with a cross-spectral stereo matching [66] method. All test images of the RGB-NIR

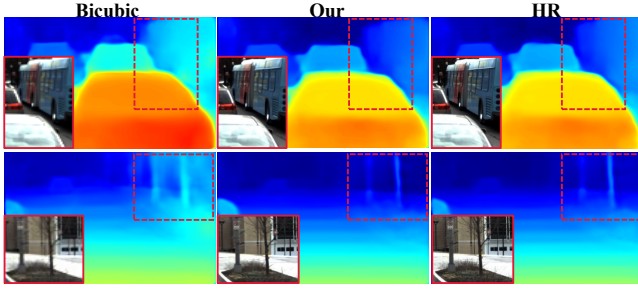

**Figure 8: The cross-spectral disparity between the SR VIS image and the HR NIR image. Our CSPSR can restore clear edge textures for accurate cross-spectral matching.**

Stereo dataset are labelled with material segments in 8 classes, including common, light, glass, glossy, vegetation, skin, clothing, and bag. By following this work, we evaluate the disparity accuracy through the disparity root mean square error (RMSE) in pixels for each material.

### 5.5 Discussion and Limitation

The main contribution of this paper is introducing cross-spectral guidance in SR of images from the multi-camera cross-spectral system. Though the near-infrared images provide benefits for the super-resolution of visible images, as we mentioned before, it may not help much for some areas with normal brightness, where the visible images contain enough high-frequency information. Therefore, how to better exploit the near-infrared image guidance for more efficient computing returns, needs to be further studied.

### 6 CONCLUSION

In this paper, we have proposed a novel cross-spectral prior guided image super-resolution (CSPSR) approach, which makes the first attempt to introduce the near-infrared (NIR) image to assist the visible image SR. The proposed CSPSR can reasonably exploit the cross-spectral guidance for recovering accurate structures and clear details through a cross-view matching (CVM) module and a dynamic multi-modal fusion (DMF) module. Specifically, the CVM module enhances the cross-view correspondence, which facilitates cross-spectral images providing more valuable and appropriate guidance for SR. The DMF module adopts a dynamic convolution and a cross-spectral feature transfer unit to adaptively enhance the multi-modal features and fully fuse the cross-spectral information for predicting abundant and realistic image details. Extensive experiments have demonstrated that the CSPSR can take full advantage of the NIR information to restore high-quality images with accurate structures and clear details, obtaining superior SR results compared to the state-of-the-art.

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
