# OpenReview forum: "Learning Cross-Spectral Prior for Image Super-Resolution"
_acmmm.org/ACMMM/2024/Conference — MM2024 Poster_

### Official Review · Reviewer_3pZx · 2024-05-20

**Rating:** 4
**Confidence:** 3

**Summary:**

This paper proposes an image super-resolution method that utilizes cross-spectral information to assist visible light images. The authors designed a cross-view matching (CVM) module and a dynamic multi-modal fusion (DMF) module to enhance the spatial correlation between cross-spectral images.

**Strengths:**

Using cross-spectral information to assist super-resolution is a relatively innovative idea. To my knowledge, no one has ever done this. The experiments in the paper are also quite sufficient. Comparing the images, we can see that the method is improved compared with the previous methods.

**Limitations:**

The data set used in the article's experiment is not commonly used in super-resolution such as Set5 and Urban100. I can understand the authors' desire to demonstrate the advantages of CSPSR on cross-spectral images. However, as a super-resolution method, it should still provide data sets other than cross-spectral images. The comparison methods in this paper are not aimed at cross-spectral images. I think the authors should provide some existing methods that work on cross-spectral images in the comparison method. This will make the experiment more convincing.

**Suitability:**

3

---

### Official Review · Reviewer_7aHf · 2024-05-22

**Rating:** 4
**Confidence:** 3

**Summary:**

This paper uses the characteristics of cross-spectral images to design a model network CSPSR. It utilizes cross-spectral and cross-view image information to restore HR images. The network enhances pixel correspondence through cross-view matching module. And the author uses dynamic convolution and a confidence-based cross-spectral feature transfer unit to build a dynamic multi-modal fusion module to fuse the features of the cross-spectral image, and then transfer the features of the NIR image to the VIS image.

**Strengths:**

The paper proposes to fuse features of cross-spectral images to guide super-resolution, which is a very interesting idea.

The experiment was very thorough.

**Limitations:**

1.Although combining cross-spectral image features is a good idea, I have not been able to see its specific effect on VIS image super-resolution from the analysis you provided. I hope you can provide a more detailed explanation.
2.The paper states that CSPSR can use the characteristics of NIR images to guide the super-resolution of VIS images, but the data in the experiment mainly uses cross-spectral images, so I have doubts about whether this network structure can have good general applicability.

**Suitability:**

3

---

### Official Review · Reviewer_rAfc · 2024-05-24

**Rating:** 4
**Confidence:** 3

**Summary:**

This paper introduces an innovative approach to image super-resolution termed Cross-Spectral Prior Guided Image Super-Resolution (CSPSR), marking the first instance where a Near-Infrared (NIR) image is utilized to enhance the super-resolution of a Visible (VIS) image. The DMF module is designed to dynamically connect the gap between different feature modalities and facilitate the transfer of knowledge across spectral domains through the use of dynamic convolution and a Cross-Spectral Feature Transfer (CSFT) unit in a dual-branch framework .

**Strengths:**

This paper is well-structured, and the English expression is good.
This paper is a work to explore cross-spectral prior guided image super-resolution (CSPSR) approach, which introduces the NIR image to assist the SR of the VIS image for the first time.
The experiments in this paper retrained six SISR methods and five Prior-guided SR methods on the RGB-NIR stereo dataset. Extensive comparative experiments demonstrate the superior performance of the methods proposed in this paper.

**Limitations:**

The experiment regarding Cross-spectral disparity estimation results on Super-Resolution (SR) images only compared the effects of Single Image Super-Resolution (SISR) methods. However, judging from the previous experiments, Prior-guided Super-Resolution methods may have better experimental outcomes. Comparing only SISR is not convincing; it lacks more reasonable comparative methods. For instance, "Learning Spatial-Spectral Prior for Super-Resolution of Hyperspectral Imagery."

Limited novelty: This paper merely designs a dynamic multi-modal fusion module based on dynamic convolution, which employs a dual-branch approach, and this is not a particularly novel practice.

**Suitability:**

2

---

### Official Review · Reviewer_f5Kp · 2024-05-24

**Rating:** 3
**Confidence:** 2

**Summary:**

The paper tries to use near-infrared (NIR) image guidance for better SR, based on the observation that NIR images can preserve valuable information for recovering adequate image details.

**Strengths:**

1. The analysis of cross-spectral SR in this paper is very detailed, pointing out some challenges, and Motivation is reasonable.
2. Compared with some algorithms, the proposed algorithm achieves better results.
3. The description of this article is relatively clear.
4. The experiments are relatively sufficient, and all parts have been tested.

**Limitations:**

The method of using cross-spectral priors was proposed a long time ago, and the use of priors lacks innovation.

**Suitability:**

3

---

### Meta-Review · Area_Chair_49FE · 2024-06-28

**Recommendation:** Accept (Poster)
**Confidence:** 4

**Metareview:**

The paper proposes a novel model network CSPSR to exploit the cross-spectral and cross-view characteristics of images to restore high-resolution image.
The paper is well written. Several minor concerns have been raised about experimentation. The authors should aim to address those in the final draft.